# Osteoarthritis and Diabetes: Where Are We and Where Should We Go?

**DOI:** 10.3390/diagnostics13081386

**Published:** 2023-04-10

**Authors:** Aqeel M. Alenazi, Ahmed S. Alhowimel, Mohammed M. Alshehri, Bader A. Alqahtani, Norah A. Alhwoaimel, Neil A. Segal, Patricia M. Kluding

**Affiliations:** 1Department of Health and Rehabilitation Sciences, Prince Sattam Bin Abdulaziz University, Alkharj 11942, Saudi Arabia; 2Departement of Physical Therapy, Jazan University, Jazan 45142, Saudi Arabia; 3Department of Physical Medicine and Rehabilitation, University of Kansas Medical Center, Kansas City, MI 66160, USA; 4Department of Physical Therapy and Rehabilitation Science, University of Kansas Medical Center, Kansas City, MI 66160, USA

**Keywords:** osteoarthritis, diabetics, high blood glucose, pain intensity, symptoms, gait, pace

## Abstract

Diabetes mellitus (DM) and osteoarthritis (OA) are chronic noncommunicable diseases that affect millions of people worldwide. OA and DM are prevalent worldwide and associated with chronic pain and disability. Evidence suggests that DM and OA coexist within the same population. The coexistence of DM in patients with OA has been linked to the development and progression of the disease. Furthermore, DM is associated with a greater degree of osteoarthritic pain. Numerous risk factors are common to both DM and OA. Age, sex, race, and metabolic diseases (e.g., obesity, hypertension, and dyslipidemia) have been identified as risk factors. These risk factors (demographics and metabolic disorder) are associated with DM or OA. Other possible factors may include sleep disorders and depression. Medications for metabolic syndromes might be related to the incidence and progression of OA, with conflicting results. Given the growing body of evidence indicating a relationship between DM and OA, it is vital to analyze, interpret, and integrate these findings. Therefore, the purpose of this review was to evaluate the evidence on the prevalence, relationship, pain, and risk factors of both DM and OA. The research was limited to knee, hip, and hand OA.

## 1. Introduction

Osteoarthritis is one of the most common chronic joint-affecting diseases. OA affects approximately 16% of the global population [1], and 26 million individuals in the United States are estimated to have OA, with the average annual cost per patient exceeding USD 2000 [2,3,4,5]. The prevalence of OA is known to increase with age, and approximately 34% of adults aged >65 years have OA [6]. OA is characterized by loss of cartilage, osteophyte formation, and synovial inflammation and the most common sites include the knee, hip, hands, and spine. Pain is the most common symptom requiring treatment. Pain severity may be influenced by many factors such as age, sex, obesity, and other comorbidities, such as DM.

Osteoarthritis can be classified based on the number of joints involved and their locations: generalized OA (GOA) affects three or more joints [7], while localized OA affects fewer than three joints [7,8]. Individuals with GOA may present with worse symptoms or poorer outcomes in terms of pain, functional impairment, and decreased quality of life. Previous evidence has shown that patients with total knee replacement and OA of multiple joints have worse pain and impaired physical function [9]. GOA affects joint replacement outcomes, quality of life, and functionality to a greater extent than localized OA [10]. Patients with GOA may have severe impairments during daily activities, that could negatively affect their self-care and basic independence [10]. Further research is needed regarding GOA in terms of prevalence, sites, and related risk factors. 

Globally, DM is one component of metabolic syndrome. DM affects approximately 11% of the general population and can lead to several complications [11]. The number of people with DM is estimated to rise to approximately 592 million globally by 2035 [12]. In the United States, more than 20 million people have DM, with a total annual cost exceeding USD 245 billion [13]. Direct out-of-pocket cost estimations range between USD 242 in Mexico to USD 11,917 in the United States [14]. A common complication of DM is hyperglycemia, which may affect joints and bones. Future research is needed in the area of cost estimation in the presence of DM and OA.

Diabetes is characterized by a disturbance in the insulin machinery that leads to hyperglycemia, which may induce chronic systemic inflammation that leads to systemic changes in the body organs and joints, and often leads to other complications [15]. Additionally, hyperglycemia can produce advanced glycation end products (AGE) that can accumulate in any part of the body, including the joints, and may increase cartilage stiffness and bone fragility [16]. 

The progression rate of DM differs depending on the associated risk factors such as demographics, presence of other chronic diseases, and poor glycemic control. Recent guidelines suggest that early treatment of DM and good glycemic control may slow DM progression [17]. In a previous study, sex and age played a significant role in poorer glycemic control, as measured by HbA1c ≥ 7; poorer glycemic control was noted in individuals who were of younger age and found to be more common in females [18]. Moreover, in the same study, other factors for poor glycemic control were also identified, including poor medication adherence, poor lifestyle modifications, and longer DM duration. Higher body mass index (BMI) and other comorbidities, such as dyslipidemia and vascular complications, were insignificant in the multivariable analyses. Additionally, another large study found that younger age was consistently associated with poor glycemic control (HbA1c ≥ 7) in patients with untreated DM [19]. Furthermore, DM can present with other comorbidities such as osteoarthritis and may result in other complications such as pain.

Diabetes mellitus (DM) and osteoarthritis (OA) are common chronic diseases that have various complications, including hyperglycemia and pain. These conditions coexist due the shared risk factors such as obesity and aging [20,21,22]. The global prevalence of OA and DM has significantly increased, respectively affecting approximately 16% and 11% of the general population [1,4,11,23]. Recent evidence has indicated an association between DM and OA [24,25]. The impact of DM was not only evident in the presence of OA, but also had a negative impact on pain and functionality in people with OA. Additionally, evidence has further shown that DM is associated with increased pain severity and decreased walking speed in patients with knee OA [26,27,28,29,30,31,32].

According to previous studies, the following shared risk factors are associated with DM and OA: demographic factors such as age, sex, and race; and metabolic syndromes such as obesity, hypertension, and dyslipidemia [20,21,22,33]. Although previous work has linked DM to OA [24], this was attributed to obesity in people with DM [34]. Previous systematic reviews and meta-analyses showed inconsistent results regarding the association between DM and OA [24,25,34,35]. Some studies included type 2 DM (T2DM) and specific joints such as knee and hand OA. Other narrative reviews have looked at the association between metabolic syndrome and OA [16,36,37,38,39,40]. Previous reviews were limited to specific joints for OA and specific forms of DM such as T2DM and did not investigate the impact of DM on symptoms and physical function among people with OA. Since DM and OA are systemic diseases with low-grade inflammation, it is crucial to look at possible joints that might be affected other than knees or hands. OA can affect knee, hip, hand, ankle, shoulder, elbow, back, and other joints. These joints were not clearly included in previous studies regarding the association of OA and DM. 

With a growing body of evidence reviewed supporting the association between DM and OA [36,37,38,41,42], to gain greater insight, it is important to review and evaluate the pertaining literature and summarize the findings on this topic. Therefore, the purpose of our review was to evaluate and review the literature on the association between DM and OA in terms of prevalence, association, symptoms, physical function, and shared risk factors. 

## 2. Methods 

All the reviewed articles were identified through an online search using PubMed, Scopus, Web of Science, Cochrane library, and Google Scholar. Keywords included “diabetes” and “osteoarthritis”. The search was limited to full-length articles of studies on humans, published in English from inception until July All search strategies for the used databases were found in Appendix A.

## 3. Results and Discussion 

### 3.1. Association between Osteoarthritis and DM

Possible mechanisms for the association between OA DM and OA have been proposed in previous reports [36,38]. The potential pathophysiology for OA is related to local and systemic low-grade inflammation [43]. The joints include the articular cartilage involving extracellular matrix that includes chondrocytes. These cells are responsible for extracellular matrix synthesis [16,37,43]. The main function of cartilage is stress and shock absorbent between the surfaces of two bones. OA indicates abnormality of these functions and a production of pro-inflammatory mediators by chondrocytes such as cytokines, tumor necrosis factors, radical oxygen species, advanced glycation end products (AGE), and prostaglandins. All of these inflammatory mediators induce an increase in proteolytic enzymes called matrix metalloproteinases (MMPs) and aggrecanases. These enzymes lead to destruction of the cartilage matrix. The presence of DM in OA status might facilitate the process joint damage through two pathways. The first pathway is via chronic hyperglycemia leading to an increase in oxidative stress and overproduction of pro-inflammatory cytokines as well as AGEs within joints. The second pathway is through insulin resistance that might have a negative impact locally and systemically through low-grade inflammation. Chondrocyte damage and apoptosis might be induced due to leptin secretion from adipose tissue leading to the increased production of cytokine and MMPs [16,37,43]. 

Numerous studies have investigated the association between DM and OA. Two meta-analyses found a significant association [24,25]. One, a large meta-analysis conducted by Louati et al. [24], that included 49 studies (study designs included cross-sectional, case–control, and cohort studies), showed that the prevalence of OA among 5788 patients with DM was 29.5% and the prevalence of DM among 645,089 patients with OA was 14.4%. This meta-analysis further revealed that the risk of OA was significantly associated with DM compared to the non-DM population, with an odds ratio (OR) with 95% confidence interval (95% CI) (OR: 1.46, 95% CI [1.08 to 1.96]. Additionally, it found that the risk of DM was significantly associated with OA compared to that in the non-OA population (OR = 1.41, 95% CI [1.21 to 1.65]). However, several studies included in this meta-analysis had limitations, including joint replacement as the main outcome and lack of control for other risk factors such as age, sex, obesity, and heterogeneous OA and DM definitions. The other meta-analysis published by Williams et al. [25] found similar results with fewer included studies (*n* = 10). This meta-analysis included studies that examined the association between OA and DM, even after controlling for BMI, in a smaller population (*n* =16,742 patients). The main outcome was the presence or progression of OA with DM as an independent factor. This meta-analysis by Williams found a significant association between OA and the presence of DM (OR = 1.21, 95% CI [1.02 to 1.41]), which remained significant after controlling for BMI. However, it also had limitations, including self-reported DM and joint replacement as the main outcomes in some included studies. 

Conversely, another meta-analysis found that the association between DM and OA was confounded by BMI [34]. This meta-analysis included 31 articles, with a pooled sample size of 295,100 individuals, and examined the bidirectional association between DM and OA, indicating that the risk of DM was higher in people with OA than in those without OA (OR:1.56, 95% CI [1.28 to 1.89]). However, the risk of OA was insignificant in people with DM compared to those without DM (OR:1.14, 95% CI [0.98 to 1,33]). The bidirectional relationship might be related to both diseases having similar features such as systemic inflammation and pro-inflammatory mediators. Regardless of the conflicting evidence between these meta-analyses, the cost of coexisting conditions could increase, especially among older adults. Recent evidence showed that the median out-of-pocket cost for common comorbidities, including DM and OA, was USD 1999 in 2019 [3]. Future systematic reviews may highlight the association between DM and OA in relation to specific joints and uniform definitions for DM and OA. 

According to one study, the prevalence of OA was estimated to be 52% in individuals with DM, compared to 27% in those without DM [44]. Numerous studies have reported a high prevalence of DM in OA populations and vice versa [45,46,47]. In a large population study (*n* = 9541), Kim et al. reported that the prevalence of knee OA was 42.4% in patients with DM, compared to 35.4% in those without DM [46]. Another small study (*n* = 202) showed that the prevalence of OA among people with DM was 49% compared to 26.5% among those without DM [45]. While a larger population-based study (*n* = 7714) reported that the prevalence of hyperglycemia was 30% in people with OA compared with 13% in those without OA [47]. These previous studies have focused on specific joint locations, such as knee joints [45,46], hands [45], or unspecified OA joints [47]. Therefore, other factors, such as BMI for weight-bearing joints, should be considered in future research analyses. Further work is needed regarding weight-bearing and non-weight-bearing OA and DM. 

Numerous reports have examined the association between DM and OA with inconsistent results within and between studies. This discrepancy might be related to mediating factors for DM and OA such as age, BMI, and sex. A previous cross-sectional study (*n* = 202) [45] showed that people with DM had 2.18 odds of having knee or hand OA compared to those without DM after adjusting for age, sex, obesity, and other risk factors (OR = 2.18, 95% CI [1.12 to 4.24]). However, this study had limitations, such as the inclusion of only Hispanic people, a small sample size, and reliance on self-reported DM. Similarly, another study found a significant association between knee OA and DM (OR = 1.19, 95% CI [1.00 to 1.41]), which was a large cross-sectional study (*n* = 9514) of Koreans [46], even after controlling for age and sex. However, after further controlling for other factors such as BMI, the association became insignificant [46]. A potential reason for this insignificant association is that DM can be categorized as either prediabetes or DM since previous meta-analysis found no association between prediabetes and OA, indicating that mild hyperglycemia might not be associated with an increased risk of OA [40]. Another possible explanation is the mediation effect of obesity on DM and OA as a shared risk factor. In a large population-based study (*n* = 7714) conducted by Puenpatom et al., the association between metabolic syndrome and OA was stronger in younger participants (mean age = 43 years) [47]. However, this study did not specify which joints were affected by OA and OA types (primary or secondary). A recent cross-sectional study from China with a large sample size (*n* = 5764) found that hyperglycemia was associated with knee OA (OR = 1.36, 95% CI [1.18 to 1.57]) in an unadjusted analysis [48]. However, this association was not observed in the age- and sex-adjusted models. Since DM and OA are affected by age, future research should focus on younger adults to better understand the relationship. 

In contrast to the previous studies discussed, a recent systematic review of 40 studies examined the association between DM and OA of the knee, hip, and hand separately [35]. This review concluded that little evidence suggests an association between DM and knee OA independent of obesity, and no evidence suggests an association between DM and hip or hand OA [35]. Consistent with this systematic review, a large case–control study (*n* = 13,500 cases; *n* = 13,500 matched controls) by Frey et al. reported that DM was not associated with hand OA, even after adjustments for age, sex, and BMI [49]. Although this study included a control group, it had some limitations. The type of hand OA or joints affected within the hand were not specified. This study used only one diagnostic code to define DM and hand OA, which may have affected accuracy, while other research used two codes to improve validity. Consistent findings from Japan (*n* = 119 women) showed that DM was not associated with knee OA [50]. However, the participants were only women who underwent knee joint surgery, indicating end-stage knee OA. However, the surgical decision might be affected by the presence of DM. Collectively, the common limitations in these studies are the focus on localized OA in specific joints, such as the knee or hand, and the cross-sectional designs.

Only a few longitudinal studies have been conducted to examine the association between DM and OA, with contradictory results. A previous study with a 12-year mean follow-up (*n* = 19,089 cases with OA.; *n* = 19,089 controls) examined the incidence of DM in patients with OA compared to those without, and found that OA was a significant risk factor for DM incidence, except for older men (>65 years), after adjusting for covariates, including obesity [51]. The OA locations and type (primary or secondary) were not specified. Another study (*n* = 927) examined the association between DM and total hip or knee replacement over a 20-year follow-up and found a significant association between DM and hip or knee replacement, after adjusting for age, sex, obesity, and other confounders [29]. This study defined OA as a total hip or knee replacement and might not be representative for the overall OA population. A previous longitudinal study (*n* = 1690) with a three-year follow-up showed an association between knee OA occurrence and DM after adjusting for confounders such as age, sex, and BMI [52]. Another large longitudinal study with a 13.5-year mean follow-up (*n* = 16,362) examined the incidence of DM among people with OA, concluding that knee and hip OA were significant predictors of DM incidents after adjusting for covariates such as age, sex, and BMI [53]. This study did not measure changes in other factors over time. In contrast to the previous study findings and concepts, a recent report (*n* = 987) examined whether DM at baseline was a predictor for radiographic knee OA over a 7-year follow-up period, and found that baseline DM was not associated with incidence of radiographic knee OA, after adjusting for confounders, including BMI [54]. However, the homeostasis model of assessment level was negatively associated with incident knee OA in women only (OR = 0.80, 95% CI [0.69 to 0.94]). Previous reports had different sample sizes, methodologies, and definitions of OA and DM. Future research should examine the longitudinal relationship between DM and OA using objective measures such as glucose level for DM and X-ray and joint symptoms for OA.

### 3.2. Progression of Osteoarthritis and Diabetes

Treatment options for OA are mainly focused on decreasing symptoms and preventing or slowing disease progression; however, DM may facilitate OA progression. Previous evidence has shown DM to be an independent risk for OA progression in addition to negative outcomes and complications following joint replacement surgery [29,52,55,56,57,58]. Schett et al. [29] evaluated arthroplasty rates among 927 patients over a 20-year follow-up. They concluded that DM was an independent risk factor for hip and knee joint replacements. Another study (*n* = 559) examined the progression of knee OA and found that DM was an independent risk factor for knee joint space narrowing, over 3 years, compared with patients without DM [55]. Another report (*n* = 1690), with a 3-year follow-up, showed that DM was associated with knee OA progression [52]. However, after further adjustment for BMI, this association was no longer observed. Another large-scale study by Nielen et al. (*n* = 94,609) found that severity of OA requiring surgery was negatively associated with increased DM severity. However, DM severity affects surgical decisions in patients with OA. The previously mentioned studies regarding DM and OA progression were predominantly focused on knee or hip OA. Moreover, OA progression has previously been linked to excessive weight-bearing stress on the joints, which could facilitate disease progression. A high mechanical load on weight-bearing joints, such as the knee or hip, may cause cartilage damage and misalignment, which may contribute to OA progression [59,60,61]. However, these studies did not examine non-weight-bearing joints. Regardless of mechanical stress, previous research found an association between OA in non-weight-bearing joints and obesity, which may suggest a systemic pathway [45,62]. The prevalence of DM in weight-bearing versus non-weight-bearing joint OA is of interest to better understand this relationship.

### 3.3. Pain and Physical Function in Osteoarthritis and Diabetes

Pain is a common symptom in patients with OA, and may be affected by DM. Pain can be categorized as nociceptive or neuropathic in people with OA [63]. Nociceptive pain occurs due to painful stimuli resulting from inflammation in the synovium and subchondral bone and is usually characterized by a sharp and/or dull aching pain. Neuropathic pain occurs due to nerve pathology and is usually described as burning, tingling, and/or numbness. For both types of pain, pain severity plays a significant role in choosing the appropriate treatment, including pharmacological intervention such as pain medications and non-pharmacological interventions such as physical therapy for pain relief. However, limited research has examined the impact of DM in the different pain categories in individuals with OA.

Numerous studies have examined the impact of DM on pain severity in people with OA. Table 1 summarizes the studies that have examined the association between DM and OA in terms of pain. Recent evidence has shown that DM is associated with increased pain severity in patients with OA [27,28,29,30,31,50,64,65,66,67,68,69,70]. A previous study (*n* = 927) found that DM was associated with more severe clinical symptoms, including pain, in patients with hip or knee OA [29]. This study included patients who underwent hip or knee arthroplasty and may have had end-stage OA. A different study (*n* = 70) concluded that patients with DM had higher pain severity in knee OA than patients without DM [27]. Moreover, this study found that the levels of inflammatory markers such as interlukin-6 and synovitis were higher in patients with DM and knee OA than in patients with only knee OA. These values were significantly associated with pain severity [27]. This study included patients who underwent arthroplasty, with a small sample size. Another work by Eitner et al., using data from the Osteoarthritis Initiative, showed that DM was associated with a worse numeric rating scale for pain and worse knee injury and osteoarthritis outcome score for pain after controlling for age, sex, BMI, and OA severity [65]. Consistent with these previous reports, Abourazzak et al. reported that DM (*n* = 130) was associated with higher pain severity in women with knee OA [28]. Additionally, another study (*n* = 119 women) showed that elevated blood glucose levels were associated with the severity of symptomatic knee OA [50]. This study included only women who were scheduled for knee joint surgery. A study conducted on 70 patients with knee OA and 81 controls found that DM was associated with higher pain severity in people with knee OA [64]. A longitudinal report by Scherzer et al. (*n* = 845) from the Johnston County Osteoarthritis Project found that DM was associated with worsening hand OA pain [67]. The majority of the studies mentioned above had the following limitations: lack of controlling for covariates, such as medications used for pain and DM.; being focused on pain at rest without examining pain during activities, as it might be a strong barrier for activities; and some studies included only a specific sex (females) in the sample. Future research is needed to examine the association between DM duration, glycemic control, and pain symptoms using different pain measures including subjective (i.e., self-reported) and objective pain testing.

Our recent evidence examined the association between DM and pain in individuals with DM after controlling for possible confounders, including medications. This study reported a significant association between DM and pain in patients with OA (*n* = 819) after controlling for pain and metabolic syndrome medications [66]. This study revealed that DM was significantly associated with increased pain severity in people with OA after adjusting for covariates including age, sex, OA locations, BMI, depression, hypertension, dyslipidemia, and medication usage (pain medications including opioids, non-opioids, and benzodiazepine; anti-diabetic; antihypertensive; antilipemic; and anti-depressants) within 90 days of the index date. Limitations in this study included the retrospective design and using diagnostic codes that might have influenced the results since misclassification bias is common in clinical settings. To improve accuracy, use of a second confirmatory code is recommended. 

Pain during activities in people with DM and knee OA was studied in our recent work. We specifically investigated and evaluated the pain while walking in patients with knee OA and DM (*n* = 1790) [31]. Our study showed that DM was significantly associated with moderate and severe pain while walking compared to no DM and no pain while walking, after controlling for age, sex, BMI, depression symptoms, and OA grade. This work has some limitations such as the key factor (DM) being self-reported without specifying type 1 or type 2, and the duration of DM was not recorded. More research is needed to investigate the impact of DM and its duration and measurement (i.e., glycemic control using A1c) on different pain symptoms during activities. 

The literature on DM’s impact on physical function (i.e., walking speed) is very limited. Our previous report examined the association between arthritis and DM and walking speed in the general population (*n* = 1255). This study found that the presence of combined arthritis and DM was associated with decreased walking speed (β = −0.11, 95% CI [−0.17, −0.6], *p* < 0.001). Another report examining the impact of DM on walking speed in people with knee OA showed similar findings (*n* = 1790) [31,72]. This study revealed that DM was significantly associated with decreased walking speed (B = −0.064; 95% CI = −0.09, −0.03) after controlling for age, sex, knee pain while walking, BMI, depressive symptoms, and OA grade. Previous work by Kendzerska et al., using a longitudinal design (*n* = 16,362) with an average of 13.5 years of follow-up, found that the number of hip/knees with OA was associated with DM incidence, and this relationship was explained by the presence of walking difficulty [53]. Although this study used a longitudinal design controlling for possible covariates such as age, sex, and BMI, walking difficulty was not quantified and it was based on a self-report. Therefore, our recent study (*n* = 4313) examined the incidence of DM among people with, or at risk of, knee OA using baseline walking speed as a predictor [73,74]. This report found a 7% cumulative incidence of DM over a 96-month follow-up period. Reduced walking speed was a predictor of incident DM (RR, 0.44; 95% CI [0.22 to 0.86] *p* = 0.018) in people with, or at risk of, knee OA. This study identified the threshold for baseline walking speed at 1.32 m/s with an area under the curve of 0.59 (*p* < 0.001), which significantly predicted DM incidence. Past research compared muscle strength in people with DM and knee OA to those with knee OA only [75]. This study found that people with DM and knee OA had lower grip strength and balance compared to those with knee OA only [75]. More work is needed regarding the impact of DM on physical functions such as gait speed, balance/mobility functions, and muscle strength in people with OA in different joints such as weight-bearing and non-weight-bearing joints. 

### 3.4. Shared Risk Factors for Osteoarthritis and Diabetes

Common risk factors associated with DM and OA are shown in Table 2 including demographic factors (age, sex, and race) and metabolic syndrome (obesity, hypertension, and dyslipidemia). Previous research found that these risk factors (demographic and metabolic syndrome) were associated with either DM or OA [20,21,22,33]. In addition to these risk factors, other factors, including medications, have been considered risk factors for either DM or OA. Recent evidence suggests that metabolic syndromes and their medications may affect the incidence and prevalence of OA [37]. Recent evidence has shown that antilipemic or antihypertensive medications are associated with decreased knee OA progression and pain [76]. Our previous research on people with OA (*n* = 3855) found that chronic diseases were associated with generalized OA compared with localized OA [22]. This study reported that the odds of generalized OA increased in people with DM (OR: 1.37, 95% CI: 1.05 to 1.78, *p* = 0.02), hypertension (OR: 1.99, CI: 1.63 to 2.43, *p* < 0.001), and dyslipidemia (OR: 3.46, CI: 2.86 to 4.19, *p* < 0.001). Demographics, including older age, female sex, race, and BMI, were associated with generalized OA compared to localized OA. 

#### 3.4.1. The Role of Age in DM and OA

Aging has a detrimental impact on different systems and organs because advanced age is associated with the decline in cellular function, that has been linked to both OA and DM [20,21,33]. Aging is a common risk factor for both OA and DM, and increased age is associated with disease development and progression. OA is associated with aging owing to a cellular decline in joints, such as chondrocytes, resulting in cartilage degradation [78]. DM is prevalent in older patients because pancreatic cell decline increases with age [77]. Age is usually controlled for during analysis investigating the association between DM and OA. However, future reports should shed light on age as one of the associated risk factors and whether age differs in people with DM, OA, or both. 

#### 3.4.2. The Role of Sex in DM and OA

The prevalence of OA is greater in women, but previous studies have usually controlled for sex in the analyses. Previous research has suggested that women have a higher prevalence of hip and knee OA than men [79,80]. A meta-analysis showed differences in the prevalence and incidence of OA based on sex, and the analysis revealed that women have an especially higher risk after menopause [90]. Conversely, a recent study reported no association between hand OA and sex [49]. Sex differences in OA prevalence might be attributed to hormonal changes in women after menopause, which could partially explain this association [91,92]. Sex influences gait parameters and muscle mass leading to differences in gait variability and joint motion [93]. 

The global prevalence of DM is similar among men and women, but women have a higher prevalence of DM than men at older ages [81]. However, the age-adjusted rate for DM in the United States was 6.6 for men and 5.9 for women in 2014 [94]. As most studies on OA and DM are controlled for age and sex, there is a critical need to evaluate this relationship in future work and to analyze the results based on sex and as a whole sample. 

#### 3.4.3. The Role of Race in DM and OA

Race is a common risk factor for both DM and OA. Previous research has reported an association between non-Hispanic African Americans and OA using a national health survey in the United States [95,96]. A similar association was observed between race and DM. A previous report showed a higher prevalence of DM among non-Hispanic African Americans compared to Hispanic Americans [97]. However, previous evidence on the association between OA and DM has not examined racial differences within both conditions because of adjusting for race. Because OA and DM are independently associated with race, future research on the association between these diseases should consider race as a potential factor.

#### 3.4.4. The Role of Obesity in DM and OA

Obesity is a shared risk factor in OA and DM, and is associated with 90% of DM [83] and OA [82]. Obesity is a systemic metabolic disease that affects body organs and joints. Impaired glucose tolerance is associated with obesity and related metabolic syndromes [98]. Obesity is typically defined as excessive body weight using many formulas, such as the body mass index (BMI). Obese people have a BMI ≥ 30, and overweight people have a BMI ≥ Recent studies have shown that obesity is a significant risk factor for knee OA after controlling for covariates, such as metabolic syndromes [46,99].

Obesity might be linked to OA, owing to the effect of weight and misalignment on joints, especially weight-bearing joints that affect joint cartilage [60,61,82]. Furthermore, previous studies have reported that obesity is associated with non-weight-bearing joints such as hand OA [52,100], suggesting that obesity might be associated with systemic metabolic dysfunction rather than mechanical dysfunction [62]. Previous research has shown an association between DM and hand OA, indicating an impact other than mechanical on lower extremities due to obesity/overweight [101]. DM was also linked to pain in erosive hand OA in either type 1 or type 2, indicating low-grade inflammation related to metabolic syndrome [71,101]. Possible mechanisms are related to oxidative stress and insulin resistance. Therefore, to better understand the relationship between obesity and OA, and obesity and DM, it is necessary to study this association in terms of weight-bearing versus non-weight-bearing joints.

#### 3.4.5. The Role of Hypertension in DM and OA

Elevated blood pressure is a common form of cardiovascular disease associated with both OA and DM. The relationship between hypertension, OA, and DM has been studied as a risk factor for OA development and progression. Prior research demonstrated that the accumulation of metabolic factors, including hypertension and DM, was associated with knee OA occurrence over a three-year period after controlling for other covariates [52]. Previous studies have also reported that hypertension is significantly associated with knee OA after controlling for covariates including BMI [46,48,102,103]. 

The proposed mechanism of hypertension as a risk factor for OA development was previously reported by Findlay [104]. Vascular impairment due to hypertension may play a role in OA development and progression. Decreased blood flow with hypertension causes subchondral ischemia, which is associated with cellular dysfunction in the joints, including osteocytes and articular cartilage [104]. The previously mentioned studies shared the common limitation of examining knee OA only. The presence of DM and OA, as well as other metabolic risk factors, including hypertension, needs further research because these metabolic syndromes are systemic diseases and may contribute to further complications.

#### 3.4.6. The Role of Dyslipidemia in DM and OA

Dyslipidemia is a form of metabolic disorder, and evidence regarding its association with DM and OA is limited owing to a lack of research. Dyslipidemia indicates disturbances in the serum levels of any form of cholesterol, including high-density lipoprotein, low-density lipoprotein, total cholesterol, or triglyceride. Prior evidence has shown that dyslipidemia is associated with knee OA after controlling for other covariates such as BMI [46,99,105]. This association might be explained by the impact of lipid profile on joint properties and increased free fatty acid. Although previous studies have demonstrated an association between dyslipidemia, DM, and OA occurrence [52,106,107], other studies have conversely reported no association between them [49,108]. These studies have focused on non-weight-bearing OA joints with different definitions of dyslipidemia, thus contributing to the conflicting results. 

#### 3.4.7. The Role of Medications in DM and OA

Medications for chronic diseases may play a role in the development and progression of OA. Medications including anti-diabetic, antilipemic, and antihypertensive drugs might be associated with OA [76,109,110,111]. A previous report found that the incidence and prevalence of OA might be affected by metabolic syndromes and their medications [37]. Recent research has demonstrated that the use of medications such as antilipemic or antihypertensives is associated with decreased knee OA progression and symptoms [76]. Additionally, one study found that individuals with DM using insulin had less osteophyte formation compared to individuals with DM who were not using insulin [109]. This could be explained by an increased synthesis of proteoglycan leading to an increased matrix synthesis and decreasing matrix breakdown [109]. Since matrix breakdown is an early and destructive feature of OA, it results in osteophyte formation and cartilage loss. Furthermore, statin use has been associated with a decreased incidence and progression of knee OA [111]. Another longitudinal study over a 10-year follow-up showed that using a high dose of statins was associated with a reduction in clinically defined OA (e.g., pain) [112]. This might be attributed to altering serum lipid levels or anti-inflammatory properties by statins [111,112]. In contrast, another study found that statin users were at an increased risk of knee OA progression compared with non-statin users [110]. These conflicting findings could be related to differences in the definitions of OA (e.g., diagnostic codes versus radiographic OA), OA location, and statin dosage. Further research is required in the context of metabolic syndrome medications and OA. A recent study extensively reviewed medication use and their association with DM and OA [38]. The authors concluded that more careful consideration is required during therapy selection for people with DM and OA to avoid safety issues [38].

Only a few studies have controlled for pain medications in the context of DM and OA association. A recent retrospective study [66] (*n* = 3855), after controlling for medication usage (pain medications including opioids, non-opioids, and benzodiazepine; anti-diabetic; antihypertensives; antilipemic; antidepressants), found that DM was significantly associated with worsening pain. Another report found that DM was associated with higher pain intensity over 7 and 30 days after controlling for covariates, including pain medications [30]. Future research should control for possible confounders that may affect pain in patients with OA and DM. 

#### 3.4.8. Other Risk Factors 

Numerous risk factors, including sleep disorders and depression, may contribute to the development of OA in individuals with DM. Hyperglycemia and OA pain are common concerns in individuals with DM and OA because glycemic control by exercise is limited, owing to pain, sleep disorders, or depression. Limited evidence has linked sleep disorders and depression to either DM [86,87] or OA [88,89]. Other factors, such as joint arthroplasty, may also have an important association with DM and OA. Evidence from our previous study found that hyperglycemia, measured by an increase in A1c, was associated with increased pain severity in people with localized OA [66]. These factors should be considered in future studies to examine the association between OA and DM.

### 3.5. Future Directions

Future studies should focus on the association between DM and OA in different forms including localized and generalized forms of OA and different locations including weight-bearing and non-weight-bearing joints. Further research should examine the association between DM and OA using objective measures including blood glucose measures and A1c of DM as well as grading of OA using X-ray and clinical symptoms, and the extent to which they are associated. Disease duration and other comorbidities should be examined further, along with their relationship to physical function and pain. The different components and dimensions of pain, such as severity and frequency, should be addressed. It is important to examine the association between DM and OA in terms of weight-bearing versus non-weight-bearing joints. Furthermore, medications should be considered in future research because previous evidence has shown conflicting results regarding their impact on either DM or OA. Finally, the possible mechanisms underlying the association between DM and OA should be studied further in future research.

## 4. Conclusions

According to the literature reviewed, DM and OA coexist and are associated with incidence and progression increasing clinical evidence for the relationship. However, some studies have suggested an insignificant association after controlling for risk factors such as age, sex, and BMI. We found that DM may increase the pain severity of knee OA.; however, limited evidence prevents us from drawing a definite conclusion. Further studies are therefore needed to elucidate whether DM increases the pain severity of OA. Moreover, we identified that common risk factors, including demographics and metabolic syndromes, may affect the association between DM and OA.; however, this too requires further research. Regarding other factors, there is conflicting evidence on whether using medication contributes positively or negatively to the association between DM and OA, and thus, extensive future research is needed to clearly ascertain the role of chronic medication use. 

## Figures and Tables

**Table 1 diagnostics-13-01386-t001:** Summary of the studies examining the association between osteoarthritis and pain in diabetic individuals.

Study	Origin	Sample Characteristics	Findings	Value
Reeuwijk et al., 2010 [69]	Netherlands	Sample size: 288 (71.2% female) Age: 66 ± 8.7 years BMI: 27.2 ± 4.5 kg/m^2^ Diagnostic criteria of O.A: radiological Pain severity: 4.8	DM was associated with pain severity after adjusting age and sex.	β = 1.2, 95% CI [0.2, 2.2]; *p* < 0.05
Abourazzak et al., 2015 [28]	Morocco	Sample size: 130 (100% female) Age: 56.7 ± 8 years BMI: 32.5 ± 2.9 kg/m^2^ Diagnostic criteria: Kallgren Lawrence grade ≥2 Pain severity: 3.6 ± 1.2	DM is associated with a higher level of pain.	OR = 3.7, 95% CI [1.5–5.9]; *p* = 0.001
Eitner et al., 2017 [27]	Germany	Sample size: 70 (56% female) Age: 71 ± 7 years BMI: 31 ± 0.7 kg/m^2^ Diagnostic criteria: radiological Pain severity: N/A	Patients with end-stage OA with DM had an 8-fold increased risk of being in the high-pain group compared with patients with end-stage OA without DM after adjusted for BMI, age, and sex.	OR = 8.2, 95% CI [2.2–30.3]; *p* = 0.002
Magnusson et al., 2017 [71]	Norway	Sample size: 96 (49% female) Age: 62.2 ± 7.4 years BMI: 26.2 ± 4 kg/m^2^ Diagnostic criteria: Radiographical Pain severity: N/T	Strong and consistent associations were observed between long term type 1 DM and increased hand pain.	β = 2.78, 95% CI [1.65–3.91]
Eymard et al., 2015 [55]	France	Sample size: 559 (70% female) Age: 62.8 years BMI: 29.8 kg/m^2^ Diagnostic criteria: radiographical Pain severity: N/T	DM was not significantly associated with worsening WOMAC subscores of pain.	201 vs. 220; *p* = 0.656
Zullig et al., 2015 [70]	USA	Sample size: 300 (9% female) Age: 61.1 ± 9 years BMI: 33.8 ± 5.2 kg/m^2^ Diagnostic criteria: radiological Pain severity: 10.2 on WOMAC	DM is associated with worsening in pain level in people with knee OA.	β = −0.6, 95% CI [−0.3, 1.4]; *p* = 0.193
Scherzer et al., 2020 [67]	USA	Sample size: 852 (67.3% female) Age: 59.5 ± 7.4 years BMI: 30.9 ± 6.5 kg/m^2^ Diagnostic criteria: radiographical Pain severity: N/A	People with DM were more likely to experience worsening pain; pain was assessed using the AUStralian CANadian Osteoarthritis Hand Index (AUSCAN).	β = −5.08, 95% CI [1.38, 18.77]
Afifi et al., 2018 [68]	Egypt	Sample size: 60 (91% female) Age: 52.8 ± 8 years BMI: 39.2 ± 9 kg/m^2^ Diagnostic criteria: Radiological Pain severity: N/A	There was a significant association of WOMAC score with DM in linear regression analysis.	β = 0.31 (*p* = 0.003)
Schett et al., 2012 [29]	Germany	Sample size: 927 Age: 67.6 ± 9.6 years BMI: 27 ± 3.9 kg/m^2^ Diagnostic criteria: Radiographical Pain severity: N/T	Pain subscales of the WOMAC and KOOS scores exhibit particularly pronounced associations with type 2 DM.	β = 91.7, 95% CI [69.4–100] β = 95.0, 95% CI [77.5–100]
Eitner et al., 2020 [65]	USA	Sample size: 2481 (61% female) Age: 65 years BMI: 31.6 kg/m^2^ Diagnostic criteria: Radiographical Pain severity: N/T	Individuals with DM had worse KOOS pain, and worse NRS pain independent of BMI, OA severity, age, and sex.	β = −4.72, 95% CI [−7.22, −2.23] β = 0.42, 95% CI [0.04, 0.80]
Alenazi et al., 2019 [66]	USA	Sample size: 819 (54.3% female) Age: 65.08 ± 9.77 years BMI: 37.7 ± 0.5 kg/m^2^ Diagnostic criteria: Osteoarthritis Initiative Pain severity: 5.3	HbA1c value was significantly associated with increased joint pain severity only after adjustments for age, gender, BMI, OA location, and pain medication.	β = 0.36, 95% CI [0.036, 0.67]; *p* = 0.029
Alenazi et al., 2020 [30]	USA	Sample size: 1319 (56.5% female) Age:61.2 ± 9.04 years BMI: 30.1 ± 4.9 kg/m^2^ Diagnostic criteria: Osteoarthritis Initiative Pain severity: 5.4	DM is significantly associated with increased knee pain severity over 7 days and 30 days after adjustment for age, gender, race, depression symptoms, composite OA score, use of medication, and knee injection.	β = 0.68, 95% CI [0.25, 1.11] β = 0.59, 95% CI [0.17, 1.01]
Alenazi et al., 2020 [31]	USA	Sample size: 1790 (56.5% female) Age: 69.6 ± 8.7 years BMI: 32.3 ± 5.09 kg/m^2^ Diagnostic criteria: Osteoarthritis Initiative Pain severity: N/A	DM is significantly associated with moderate and severe pain while walking when compared with no DM and no pain while walking, and after controlling for age, gender, BMI, depression symptoms, and OA grade.	OR = 1.78; 95% CI = 1.02–3.10 OR = 2.52; 95% CI = 1.01–6.28

**Table 2 diagnostics-13-01386-t002:** Summary of shared risk factors for OA and diabetes.

Risk Factors	OA	Diabetes
Age	Older age increased the risk [77]	Older age increased the risk [78]
Gender	Females have higher risk than males [79,80]	Females have higher prevalence of DM than men at older ages [81]
Obesity	Obesity increased the risk [59,82]	Obesity increased the risk [83]
Hypertension	Associated with increased risk [52]	Associated with increased risk [84]
Dyslipidemia	Associated with increased risk [52]	Associated with increased risk [85]
Other risk factors	Previous injury, joint arthroplasty, sleep disorders [86], and depression [87]	Depression [88] and sleep disorders [89]

## Data Availability

Not applicable.

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
