# Peer review of "Osteoarthritis and Diabetes: Where Are We and Where Should We Go?"

_diagnostics, 2023, doi:10.3390/diagnostics13081386_

Round 1

Reviewer 1 Report

This article reviewed the prevalence, association, pain, and risk factors of diabetes with osteoarthritis of the knee, hip, and hand. This work is of interest, however, several issues should be addressed clearly.

1. A major problem with this paper was that it does not described the underlying mechanism associated with osteoarthritis in diabetes. It was reported in this paper that the level of inflammatory markers in patients with diabetes combined with knee osteoarthritis is higher than that in patients with knee osteoarthritis alone, which can partially explain the relationship between diabetes and osteoarthritis pain. More such analysis should be added in other parts of the article. 2. It was mentioned in the conclusions that “DM and OA coexist and are associated with incidence and progression increasing clinical evidence for the relationship” .However, some research suggested the association between DM and OA became insignificant after controlling for some common risk factors of diabetes and osteoarthritis, such as BMI, age, and sex (line167, line 176, line 181), so this should be stated in the conclusions. 3. As mentioned in the article, hyperglycemia can produce advanced glycation end products (AGE) that can accumulate in any part of the body, including the joints, and may increase cartilage stiffness 100 and bone fragility”(line 99). Please further describe the relationship between cartilage and bone changes and osteoarthritis for better understanding 4. As mentioned in the article, Previous research has reported an association between non-Hispanic African Americans and OA using a national health survey in the United States"(line 374), please indicate whether this association is consistent with the results of diabetes. 5. The article mentioned that "obesity is associated with non-weight-bearing joints such as hand OA, suggesting that obesity might be associated with systemic metabolic dysfunction rather than mechanical dysfunction "(line 391)As an important part of metabolic syndrome, diabetes and hyperlipidemia are related to many metabolic mechanisms. Is there any metabolic connection between DM and hand OA? Please comment.  6. If possible, consider adding more information to Table 1 such as the location and severity of osteoarthritis. 7. Search strategies for different databases may be added to facilitate readers’ understanding. 8. Line22: “disorder should be “disorders” 9. Line 368: “are” should be deleted.

Author Response

Dear Reviewers,

Thank you for your effort in revising our paper. The suggestions offered were immensely helpful and made this work better.

We also appreciate your insightful comments and have revised the manuscript accordingly. All revisions were highlighted in boldface across the manuscript.

Kind Regards,

Authors

Reviewer1

This article reviewed the prevalence, association, pain, and risk factors of diabetes with osteoarthritis of the knee, hip, and hand. This work is of interest, however, several issues should be addressed clearly.

  1. A major problem with this paper was that it does not described the underlying mechanism associated with osteoarthritis in diabetes. It was reported in this paper that “the level of inflammatory markers in patients with diabetes combined with knee osteoarthritis is higher than that in patients with knee osteoarthritis alone, which can partially explain the relationship between diabetes and osteoarthritis pain. More such analysis should be added in other parts of the article.

RESPONSE: Thank you for your valuable feedback. We added a paragraph to be the first in the association between OA and DM section 3.1 related to the potential mechanisms of OA and DM.

  1. It was mentioned in the conclusions that “DM and OA coexist and are associated with incidence and progression increasing clinical evidence for the relationship” .However, some research suggested the association between DM and OA became insignificant after controlling for some common risk factors of diabetes and osteoarthritis, such as BMI, age, and sex (line167, line 176, line 181), so this should be stated in the conclusions.

RESPONSE: Thank you for your feedback, we revised accordingly and added this sentence to the conclusion to clarify the conflict in the literature.

  1. As mentioned in the article, “hyperglycemia can produce advanced glycation end products (AGE) that can accumulate in any part of the body, including the joints, and may increase cartilage stiffness 100 and bone fragility”(line 99). Please further describe the relationship between cartilage and bone changes and osteoarthritis for better understanding

RESPONSE: Since we moved this section into the introduction, we explained potential mechanism for cartilage damage in OA and then in OA and DM in the first paragraph of the results section 3.1

  1. As mentioned in the article, “Previous research has reported an association between non-Hispanic African Americans and OA using a national health survey in the United States"(line 374), please indicate whether this association is consistent with the results of diabetes.

RESPONE: Unfortunately, previous evidence on the association between OA and DM has not examined racial differences within both conditions because of adjusting for the race. We added this to the paragraph to clarify.

  1. The article mentioned that "obesity is associated with non-weight-bearing joints such as hand OA, suggesting that obesity might be associated with systemic metabolic dysfunction rather than mechanical dysfunction "(line 391). As an important part of metabolic syndrome, diabetes and hyperlipidemia are related to many metabolic mechanisms. Is there any metabolic connection between DM and hand OA? Please comment. 

RESPONSE: Yes, previous research has shown an association between DM and hand OA indicating an impact other than the mechanical on lower extremities due to obesity/overweight. DM was linked to pain in erosive hand OA in either type 1 or type 2, indicating low grade inflammation related to metabolic syndrome. Possible mechanisms are related to oxidative stress and insulin resistance. We added this information to the paragraph.

  1. If possible, consider adding more information to Table 1 such as the location and severity of osteoarthritis.

RESPONSE: We revised accordingly if the studies reported these information. We also added information regarding age, BMI and other factors as requested by reviewer 2.

  1. Search strategies for different databases may be added to facilitate readers’ understanding.

RESPONSE: An additional database added to the methods:

All reviewed articles were identified through online search using PubMed, Scopus, Web of Science, Cochrane library and Google Scholar.

  1. Line22: “disorder” should be “disorders”. 

RESPONSE: Changed to be read as: disorders

  1. Line 368: “are” should be deleted.

RESPONSE: Changed to be read as: Because OA and DM independently associated with …

Reviewer 2

The review discussed the potential relationship between OA and DM. Despite the understanding that the authors want to focus on human studies, some interpretation of the findings on biological mechanism will help the readers to understand the review better.

RESPONSE: Thank you for your valuable feedback. We added a paragraph to be the first in the association between OA and DM section 3.1 related to the potential mechanisms for OA and DM.

Methods: Although the authors disclosed the keywords used for literature search, the readers will understand the search better if the authors listed the search string with Boolean operators used in the search engines. For example, diabetes AND (osteoarthritis OR join pain) AND relationship.

Response: All search strategies for used databases were added to the supplementary material 1. 

I think the basic information of OA and DM should be be part of the results. Instead, please condense them and put them in the introduction. The results should focus on the relationship between these two conditions.

RESPONSE: We moved these sections to the introduction and keep the results focused on OA and DM

Line 138: typo: ....

RESPONSE: Revised

bidirectional association between DM and OA, indicating that... As I mentioned earlier, most of the time, the authors just stated the observation without leading the readers to understand the phenomenon. In this instances: line 136-140: why is there a bidirectional relationship between OA and DM. Any biological explanation provided?

RESPONSE: Because both diseases have similar features such as systemic inflammation and pro-inflammatory mediators. This might lead to bidirectionality of the association. We added this to the paragraph.

Line 160: The use of unadjusted and adjusted analysis should not form the basis of the paragraph. If the relationship is lost after adjusting for a certain confounders, it may indicate a mediation effects. The discussion should be based on the mediating factors.

RESPONSE: This paragraph was revised, and we added mediating factors to the beginning for clarity.

Line 168: A potential reason for this insignificant association is that DM can be categorized as either prediabetes or DM. The authors need to explain further how misclassification affect the relationship.

RESPONSE: As previous research found that prediabetes was not associated with OA. We added this to clarify our explanation.

Table 1 is rather simplistic. Some important info of the studies are not included, such as age, BMI, diagnostic criteria of OA and DM etc. The findings and values column should merged or else it is difficult to judge which values refer to which statement.

RESPONSE: Thank you for your feedback. We added age, BMI, diagnostic criteria and other important information if they reported by previous studies to table1.

Table 2. Other risk factors for DM, the statement starts with ref 106. Is there a missing word?

RESPONSE: we revised this accordingly

Line 367-368 does not explain the statement before it.

RESPONE: we removed this statement for clarity.

What is the biological reason elderly women have increased prevalence of OA? Does menopause plays a role?

RESPONSE: Yes in addition to biomechanical factors such as joint mechanics and muscles. We added this to the end of the paragraph.

For all other factors listed (3.6.1 to 3.6.7), what are the biological explaination for the relationship?

RESPONSE: we revised and added some information related to biological explanations

Line 430: how does reduced osteophytes in insulin users affect the realtionship?

RESPONSE: This could be explained by increased synthesis of proteoglycan leading to increased matrix synthesis. We added this to the paragraph.

Line 433: how does statins use reduced OA features?

RESPONSE: thank you for your feedback. We revised and added the potential mechanism by lowering s

Reviewer 2 Report

The review discussed the potential relationship between OA and DM. Despite the understanding that the authors want to focus on human studies, some interpretation of the findings on biological mechanism will help the readers to understand the review better. Methods: Although the authors disclosed the keywords used for literature search, the readers will understand the search better if the authors listed the search string with Boolean operators used in the search engines. For example, diabetes AND (osteoarthritis OR join pain) AND relationship. I think the basic information of OA and DM should be be part of the results. Instead, please condense them and put them in the introduction. The results should focus on the relationship between these two conditions. Line 138: typo: .... bidirectional association between DM and OA, indicating that... As I mentioned earlier, most of the time, the authors just stated the observation without leading the readers to understand the phenomenon. In this instances: line 136-140: why is there a bidirectional relationship between OA and DM. Any biological explanation provided? Line 160: The use of unadjusted and adjusted analysis should not form the basis of the paragraph. If the relationship is lost after adjusting for a certain confounders, it may indicate a mediation effects. The discussion should be based on the mediating factors. Line 168: A potential reason for this insignificant association is that DM can be categorized as either prediabetes or DM. The authors need to explain further how misclassification affect the relationship. Table 1 is rather simplistic. Some important info of the studies are not included, such as age, BMI, diagnostic criteria of OA and DM etc. The findings and values column should merged or else it is difficult to judge which values refer to which statement. Table 2. Other risk factors for DM, the statement starts with ref 106. Is there a missing word? Line 367-368 does not explain the statement before it. What is the biological reason elderly women have increased prevalence of OA? Does menopause plays a role? For all other factors listed (3.6.1 to 3.6.7), what are the biological explaination for the relationship? Line 430: how does reduced osteophytes in insulin users affect the realtionship? Line 433: how does statins use reduced OA features?

Author Response

Dear Reviewers,

Thank you for your effort in revising our paper. The suggestions offered were immensely helpful and made this work better.

We also appreciate your insightful comments and have revised the manuscript accordingly. All revisions were highlighted in boldface across the manuscript.

Kind Regards,

Authors

Reviewer1

This article reviewed the prevalence, association, pain, and risk factors of diabetes with osteoarthritis of the knee, hip, and hand. This work is of interest, however, several issues should be addressed clearly.

  1. A major problem with this paper was that it does not described the underlying mechanism associated with osteoarthritis in diabetes. It was reported in this paper that “the level of inflammatory markers in patients with diabetes combined with knee osteoarthritis is higher than that in patients with knee osteoarthritis alone, which can partially explain the relationship between diabetes and osteoarthritis pain. More such analysis should be added in other parts of the article.

RESPONSE: Thank you for your valuable feedback. We added a paragraph to be the first in the association between OA and DM section 3.1 related to the potential mechanisms of OA and DM.

  1. It was mentioned in the conclusions that “DM and OA coexist and are associated with incidence and progression increasing clinical evidence for the relationship” .However, some research suggested the association between DM and OA became insignificant after controlling for some common risk factors of diabetes and osteoarthritis, such as BMI, age, and sex (line167, line 176, line 181), so this should be stated in the conclusions.

RESPONSE: Thank you for your feedback, we revised accordingly and added this sentence to the conclusion to clarify the conflict in the literature.

  1. As mentioned in the article, “hyperglycemia can produce advanced glycation end products (AGE) that can accumulate in any part of the body, including the joints, and may increase cartilage stiffness 100 and bone fragility”(line 99). Please further describe the relationship between cartilage and bone changes and osteoarthritis for better understanding

RESPONSE: Since we moved this section into the introduction, we explained potential mechanism for cartilage damage in OA and then in OA and DM in the first paragraph of the results section 3.1

  1. As mentioned in the article, “Previous research has reported an association between non-Hispanic African Americans and OA using a national health survey in the United States"(line 374), please indicate whether this association is consistent with the results of diabetes.

RESPONE: Unfortunately, previous evidence on the association between OA and DM has not examined racial differences within both conditions because of adjusting for the race. We added this to the paragraph to clarify.

  1. The article mentioned that "obesity is associated with non-weight-bearing joints such as hand OA, suggesting that obesity might be associated with systemic metabolic dysfunction rather than mechanical dysfunction "(line 391). As an important part of metabolic syndrome, diabetes and hyperlipidemia are related to many metabolic mechanisms. Is there any metabolic connection between DM and hand OA? Please comment. 

RESPONSE: Yes, previous research has shown an association between DM and hand OA indicating an impact other than the mechanical on lower extremities due to obesity/overweight. DM was linked to pain in erosive hand OA in either type 1 or type 2, indicating low grade inflammation related to metabolic syndrome. Possible mechanisms are related to oxidative stress and insulin resistance. We added this information to the paragraph.

  1. If possible, consider adding more information to Table 1 such as the location and severity of osteoarthritis.

RESPONSE: We revised accordingly if the studies reported these information. We also added information regarding age, BMI and other factors as requested by reviewer 2.

  1. Search strategies for different databases may be added to facilitate readers’ understanding.

RESPONSE: An additional database added to the methods:

All reviewed articles were identified through online search using PubMed, Scopus, Web of Science, Cochrane library and Google Scholar.

  1. Line22: “disorder” should be “disorders”. 

RESPONSE: Changed to be read as: disorders

  1. Line 368: “are” should be deleted.

RESPONSE: Changed to be read as: Because OA and DM independently associated with …

Reviewer 2

The review discussed the potential relationship between OA and DM. Despite the understanding that the authors want to focus on human studies, some interpretation of the findings on biological mechanism will help the readers to understand the review better.

RESPONSE: Thank you for your valuable feedback. We added a paragraph to be the first in the association between OA and DM section 3.1 related to the potential mechanisms for OA and DM.

Methods: Although the authors disclosed the keywords used for literature search, the readers will understand the search better if the authors listed the search string with Boolean operators used in the search engines. For example, diabetes AND (osteoarthritis OR join pain) AND relationship.

Response: All search strategies for used databases were added to the supplementary material 1. 

I think the basic information of OA and DM should be be part of the results. Instead, please condense them and put them in the introduction. The results should focus on the relationship between these two conditions.

RESPONSE: We moved these sections to the introduction and keep the results focused on OA and DM

Line 138: typo: ....

RESPONSE: Revised

bidirectional association between DM and OA, indicating that... As I mentioned earlier, most of the time, the authors just stated the observation without leading the readers to understand the phenomenon. In this instances: line 136-140: why is there a bidirectional relationship between OA and DM. Any biological explanation provided?

RESPONSE: Because both diseases have similar features such as systemic inflammation and pro-inflammatory mediators. This might lead to bidirectionality of the association. We added this to the paragraph.

Line 160: The use of unadjusted and adjusted analysis should not form the basis of the paragraph. If the relationship is lost after adjusting for a certain confounders, it may indicate a mediation effects. The discussion should be based on the mediating factors.

RESPONSE: This paragraph was revised, and we added mediating factors to the beginning for clarity.

Line 168: A potential reason for this insignificant association is that DM can be categorized as either prediabetes or DM. The authors need to explain further how misclassification affect the relationship.

RESPONSE: As previous research found that prediabetes was not associated with OA. We added this to clarify our explanation.

Table 1 is rather simplistic. Some important info of the studies are not included, such as age, BMI, diagnostic criteria of OA and DM etc. The findings and values column should merged or else it is difficult to judge which values refer to which statement.

RESPONSE: Thank you for your feedback. We added age, BMI, diagnostic criteria and other important information if they reported by previous studies to table1.

Table 2. Other risk factors for DM, the statement starts with ref 106. Is there a missing word?

RESPONSE: we revised this accordingly

Line 367-368 does not explain the statement before it.

RESPONE: we removed this statement for clarity.

What is the biological reason elderly women have increased prevalence of OA? Does menopause plays a role?

RESPONSE: Yes in addition to biomechanical factors such as joint mechanics and muscles. We added this to the end of the paragraph.

For all other factors listed (3.6.1 to 3.6.7), what are the biological explaination for the relationship?

RESPONSE: we revised and added some information related to biological explanations

Line 430: how does reduced osteophytes in insulin users affect the realtionship?

RESPONSE: This could be explained by increased synthesis of proteoglycan leading to increased matrix synthesis. We added this to the paragraph.

Line 433: how does statins use reduced OA features?

RESPONSE: thank you for your feedback. We revised and added the potential mechanism by lowering serum lipid levels or anti-inflammatory properties to the paragraph.

Reviewer 3 Report

Well done, no comments.

Author Response

Thank you 

Round 2

Reviewer 2 Report

Methods: I think combining all 4 concepts could be quite restrictive. I suggest using only “diabetes” and “OA” and filtering out the epidemiology studies. It is still not sure how articles from all databases are integrated and screened.

Line 121: These cells are responsible for extracellular matrix synthesis [ref]. Citation needed.

Please provide the 95% CI values for OR, HR, and RR values stated in the review (line 142, 187, 199 etc)

Line 319: the use of diagnostic code that might influence the results – are the authors implying misclassification bias?

Line 463: Increased synthesis of matrix synthesis due to anabolic action of insulin?

Table 1:

·        Please indicate in the table that the values in [ ] brackets are 95% CI.

·        For sample characteristics: diagnostic criteria, Kellgren and Lawrence system is also based on radiographic evidence. So, what is the difference between “radiography” and “Kellgren and Lawrence system”?

·        Please assign units for age and BMI.

·        For Afifi et al. why the association coefficient was not reported?

Author Response

Dear Reviewers,

Thank you for your effort in revising our paper. The suggestions offered were immensely helpful and made this work better. We also appreciate your insightful comments and have revised the manuscript accordingly. All revisions were highlighted in boldface across the manuscript.

Kind Regards,

Authors

Reviewer 2 comments

  1. Methods: I think combining all 4 concepts could be quite restrictive. I suggest using only “diabetes” and “OA” and filtering out the epidemiology studies. It is still not sure how articles from all databases are integrated and screened.

Response: Thank you for your comments. Initially, we screened and reviewed based on diabetes and osteoarthritis keywords only and it included all the included articles. We revised accordingly.

  1. Line 121: These cells are responsible for extracellular matrix synthesis [ref]. Citation needed.

Response: Thank you for your comment. We missed this during the first round of revision. It is added now

  1. Please provide the 95% CI values for OR, HR, and RR values stated in the review (line 142, 187, 199 etc)

Response: We revised and added accordingly

  1. Line 319: the use of diagnostic code that might influence the results – are the authors implying misclassification bias?

Response: Yes, it is common in clinical settings to misclassified coding for diseases. To improve accuracy, using a second confirmatory code is recommended. We added this to the sentence for clarity.

  1. Line 463: Increased synthesis of matrix synthesis due to anabolic action of insulin?

Response: thank you for your comment, we revised accordingly for clarity.

  1. Table 1: Please indicate in the table that the values in [ ] brackets are 95% CI.

Response: We added this to table 1.

  1. For sample characteristics: diagnostic criteria, Kellgren and Lawrence system is also based on radiographic evidence. So, what is the difference between “radiography” and “Kellgren and Lawrence system”?

Response: They are the same, but we reported based on the reports from the original studies.

  1. Please assign units for age and BMI.

Response: We added this.

  1. For Afifi et al. why the association coefficient was not reported?

Response: We added coefficient to the table 1